# The evolution of metastatic upper tract urothelial carcinoma through genomic-transcriptomic and single-cell protein markers analysis

Kentaro Ohara[1,2], André Figueiredo Rendeiro [2,3,4,5], Bhavneet Bhinder[2,3,4], Kenneth Wha Eng[2,4], Hiranmayi Ravichandran [2], Duy Nguyen[6], David Pisapia[1,2], Aram Vosoughi [1,2], Evan Fernandez[2,4], Kyrillus S. Shohdy[6], Jyothi Manohar[2], Shaham Beg [1,2], David Wilkes [2], Brian D. Robinson [1,2], Francesca Khani [1,2], Rohan Bareja[2,3,4], Scott T. Tagawa [2,6,7], Madhu M. Ouseph[1], Andrea Sboner [1,2,4], Olivier Elemento [2,3,4,7], Bishoy M. Faltas [2,6,7,8,9] ✉ & Juan Miguel Mosquera [1,2,9] ✉

The molecular characteristics of metastatic upper tract urothelial carcinoma (UTUC) are not well understood, and there is a lack of knowledge regarding the genomic and transcriptomic differences between primary and metastatic UTUC. To address these gaps, we integrate whole-exome sequencing, RNA sequencing, and Imaging Mass Cytometry using lanthanide metal-conjugated antibodies of 44 tumor samples from 28 patients with high-grade primary and metastatic UTUC. We perform a spatially-resolved single-cell analysis of cancer, immune, and stromal cells to understand the evolution of primary to metastatic UTUC. We discover that actionable genomic alterations are frequently discordant between primary and metastatic UTUC tumors in the same patient. In contrast, molecular subtype membership and immune depletion signature are stable across primary and matched metastatic UTUC. Molecular and immune subtypes are consistent between bulk RNA-sequencing and mass cytometry of protein markers from 340,798 single cells. Molecular subtypes at the single-cell level are highly conserved between primary and metastatic UTUC tumors within the same patient.

Upper tract urothelial carcinoma (UTUC) is defined as urothelial carcinoma (UC) arising from the ureter or the renal pelvis. UTUC is a rare tumor type compared to UC of the bladder (UCB), accounting for 5–10% of all UC tumors, with aggressive clinical presentation. While the 5-year survival rate for non-muscle invasive UTUC is more than 90%, it drops to less than 40% in patients with regional nodal metastases and <10% in patients with distant metastases[1,2]. Despite systemic therapies, most patients with metastatic UTUC die from the disease[3]. A better understanding of the underlying molecular basis of metastatic UTUC is needed to develop targeted therapeutic strategies and improve clinical outcomes for patients with metastatic UTUC.

Recent next-generation sequencing (NGS) studies have detailed the genomic landscape of primary UTUC[4–6]. Our group recently demonstrated that primary UTUC has a luminal-papillary T-cell depleted phenotype and a lower total mutational burden than UCB[7]. However, previous studies on UTUC, including ours, mainly focused

on primary tumors. Only limited molecular data currently exists from patients with metastatic UTUC[6]. Furthermore, the extent of intra-patient genomic and phenotypic heterogeneity within an individual patient with metastatic UTUC is still unknown.

In this study, we performed whole-exome sequencing (WES), RNA-sequencing (RNA-seq), and multiplexed imaging cytometry which leverages antibodies conjugated to rare metals and detection by mass spectrometry to provide spatial protein expression patterns at a single cell resolution[8,9] from patients with primary and metastatic UTUC enrolled in our precision medicine study to characterize the genomic, transcriptomic and immunophenotypic features of metastatic UTUC, and understand the degree of heterogeneity between primary and metastatic UTUC.

## Results

### Genomic landscape of metastatic UTUC

To define the genomic landscape of metastatic UTUC, we performed WES of 44 prospectively collected UTUC tumors from a new cohort of 28 patients, including 7 matched sets of primary and metastatic UTUC and germline samples and one rapid autopsy (Figs. 1a, b and Supplementary Table 1). Metastatic lesions showed an average of 196 non-synonymous Single Nucleotide Variants (SNVs)/Indels (insertions or deletions) (range 5 − 713) and 149 copy number alterations (CNAs) (amplifications and deletions, range 4−831). All the analyzed UTUC had tumor mutational burden (TMB) scores below the UC-specific threshold we previously defined[10] to designate TMB-high tumors. There were no significant differences in the MSIsensor scores[11] or TMB between primary and metastatic UTUC (Mann-Whitney U-test $P$ value = 0.65 and 0.22, respectively) (Supplementary Fig. 1). The most frequently altered genes in our UTUC cohort were *TP53* (45.2%), *KMT2D* (35.7%), *ARID1A* (33.3%), and *CDKN2A* (33.3%) (Fig. 2a). The frequently altered genes in our cohort were in agreement with previously reported primary UTUC genomic studies[4,5,7,12], except for *FGFR3* (4/42, 9.5%). Of the detected CNAs, the frequency of *RAF1* amplification was significantly higher in metastatic UTUC (33.3%) compared to primary UTUC (4.2%) (Fisher exact test $P$ value = 0.03) (Fig. 2b). We searched our dataset for tumor samples interrogated by targeted sequencing panels validated for clinical use. We found two samples with genomic data from both WES and targeted panels. One sample (WCM081_P) was tested by Oncomine and another (WCM057_M1) by 50 gene custom panel that interrogates 2800 hotspots/variants of 50 cancer-related genes. Oncomine and the 50 gene panel confirmed the druggable *FGFR3* S249C and *PIK3CA* E545K mutations, respectively.

### Genomic heterogeneity between primary and matched metastatic UTUC

To compare the clonal structure of primary and metastatic UTUC, we investigated the number of private and shared mutations between the primary and metastatic UTUC tumors within each patient. We calculated the percentage of non-synonymous mutations that were private to either the primary tumor, the metastatic tumor, or shared within an individual patient. Of the 7 analyzed patients with paired primary and metastatic UTUC, 5 received chemotherapy treatment before sampling the primary or metastatic tumor tissue. All the tumors from two patients, WCM052 and WCM068, were chemotherapy-naïve (Supplementary Fig. 2). On average, only 17.6% (range 7.3−36.4%) of mutations were shared by primary and matched metastatic samples (Fig. 3a). The percentages of shared mutations were higher in the chemotherapy naïve patients than in patients who received prior chemotherapy treatment (32.6% vs. 11.5%), suggesting that chemotherapy potentially increases genomic heterogeneity by inducing mutations[13–15].

To determine the biological impact of genomic heterogeneity, we next sought to define the differences in mutations and CNAs affecting cancer-associated genes between primary and matched metastatic

UTUC within each patient. A median of 13.5 cancer gene alterations per sample was identified (range 2 − 51), including a median of 2 (range 1−9) alterations predicted to be likely oncogenic by the OncoKB database[16]. When we compared genomic alterations between paired primary and metastatic tumors, we observed significant intra-patient heterogeneity in alterations affecting critical cancer genes (Fig. 3b). In all cases, the primary and matched metastatic tumors shared at least one pathogenic cancer gene alteration (Supplementary Fig. 3). However, compared to primary UTUC, we identified additional pathogenic mutations or CNAs in all but one of the matched metastatic UTUC tumors (Fig. 3b, Supplementary Fig. 3).

### Stability of molecular subtype and immune-contexture assignments between primary and metastatic UTUC tumors

We then evaluated the transcriptomic differences between the primary and matched metastatic UTUC using RNA sequencing of 17 UTUC tumors (six primary and eleven metastatic tumors). The cohort included three patients with primary and matched metastatic UTUC and one with two metastatic tumors (Fig. 4a). Using the recently published single-sample consensus molecular classifier[17], we found that 83.3% of primary tumor samples were luminal-papillary, and the rest were basal/squamous (Fig. 4a). Of the metastatic tumors, 45.5%, 27.3%, 9.1% and 18.2% were luminal-papillary, luminal-unstable, stroma-rich and basal/squamous, respectively (Fig. 4a). Clustering analysis using the BASE47 gene classifier[18] showed a similar frequency of basal subtype between primary and metastatic tumors (33.3% vs. 27.3%, $P$ value > 0.05). When we evaluated the molecular subtypes between primary and matched metastatic UTUC tumors, we found that they were relatively stable compared to genomic changes. Still, discordance in molecular subtype was observed in one out of three cases (Fig. 4d). In WCM010, one of three lymph node metastases was classified as luminal-unstable, while the primary tumor and the remaining lymph node metastases were classified as luminal-papillary. For patient WCM031, only the metastatic tumors were interrogated by RNA-seq due to the unavailability of frozen tissue from the primary tumor. Interestingly, molecular subtyping revealed discordance of molecular subtypes between the asynchronous liver metastases (Fig. 4b). The initial metastasis (M1) and the second metastasis (M2) demonstrated stroma-rich and luminal unstable, respectively. GSEA using RNA-seq data revealed no significant enrichment of biological gene sets between primary and metastasis (FDR > 0.25) (Supplementary Fig. 4).

Next, we applied a classifier of 170 immune-related genes, which we previously developed[7] (Supplementary Table 2), to the RNA-seq data obtained from 17 samples (6 primary and 11 metastatic UTUC) and the TCGA-BLCA cohort. The majority of UTUC tumors (76.5%, 13/17) clustered into the T-cell depleted subgroup (Fig. 4c, d), consistent with our previous findings in primary UTUC[7]. Only 16.7% of primary UTUC (1/6) and 27.3% of metastatic UTUC (3/11) were T-cell inflamed. The immune phenotypes were concordant within individual patients. Additionally, we employed CIBERSORT to estimate the relative abundance of 22 immune cell types from RNA-seq data[19]. CIBERSORT showed no significant differences in the infiltration of 22 immune cell types between primary and metastatic UTUCs. In terms of T-cell subtypes, activated memory CD4$^+$ T cells and Gamma Delta T cells were low compared to other T cell types (Supplementary Fig. 5).

### Single-cell spatial profiling of the immune contexture of UTUC reveals intra-tumoral plasticity

To investigate the spatial landscape of the microenvironment of primary and metastatic UTUC tumors, we employed Imaging Mass Cytometry™ (IMC™). This multiplexed tissue imaging method uses antibodies conjugated to rare metals detectable by mass spectrometry[20]. In total, from 14 samples of 6 patients, we generated 58 images containing the spatial distribution of 27 distinct molecular biomarkers, which included markers of tumor cells (pan-Keratin,

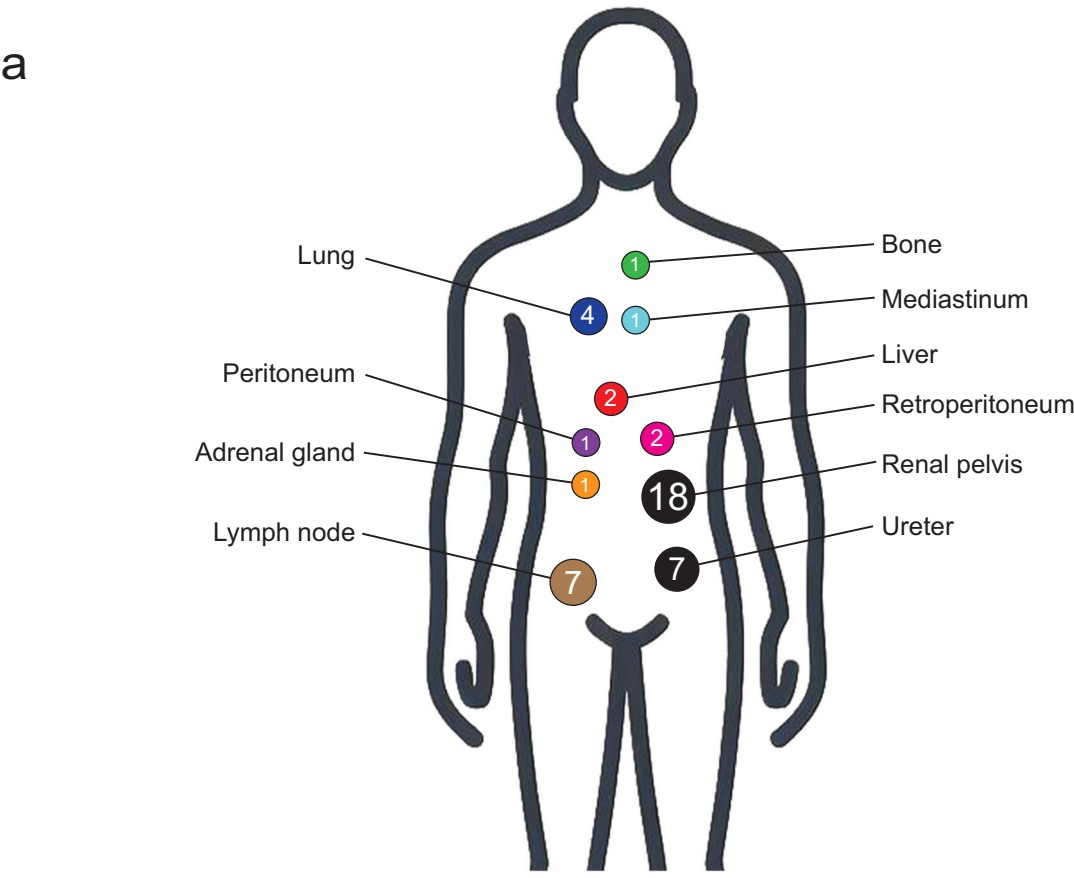

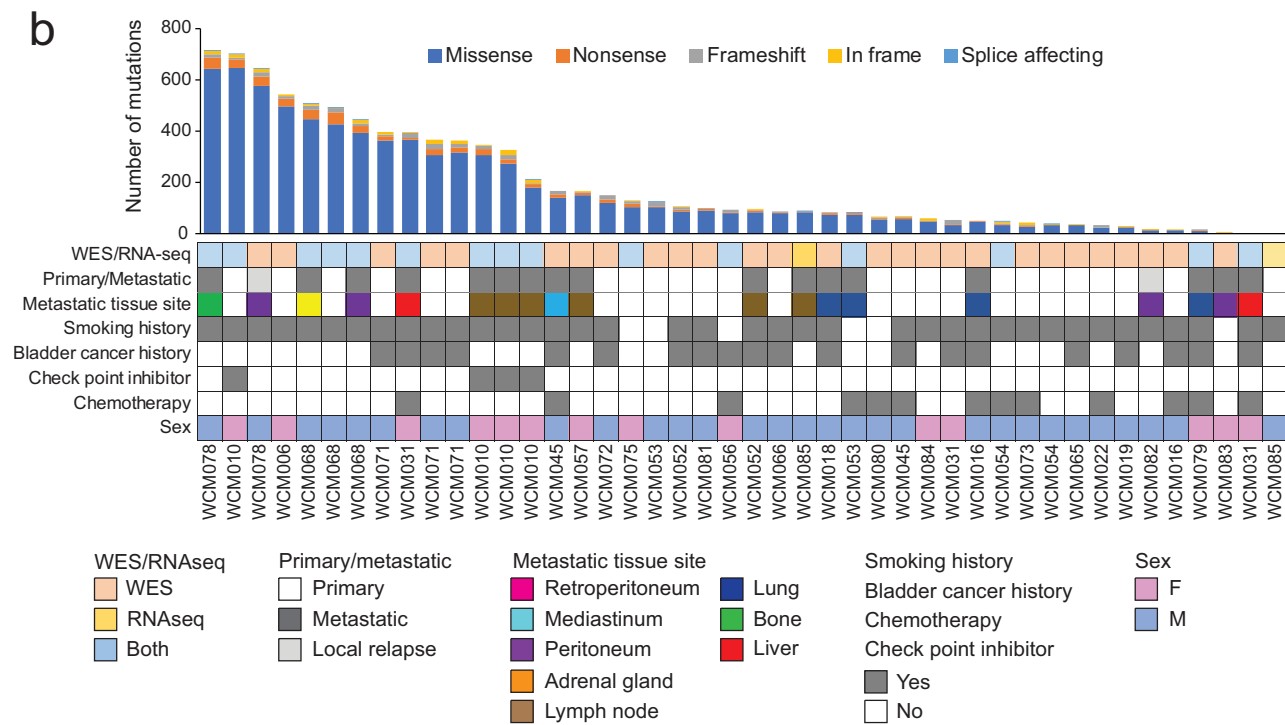

**Fig. 1 | Clinical characteristics of the study cohort. a** Schematic illustrating the anatomical sites of primary and metastatic upper tract urothelial carcinoma (UTUC) samples. Numbers correspond to the number of tumors at each site. **b** Clinical characteristics of the study cohort, the anatomical sites of primary and metastatic tumor samples and sequencing methods performed for each sample. Barplots represent numbers of non-synonymous mutations for each sample. Source data are provided as a Source Data file.

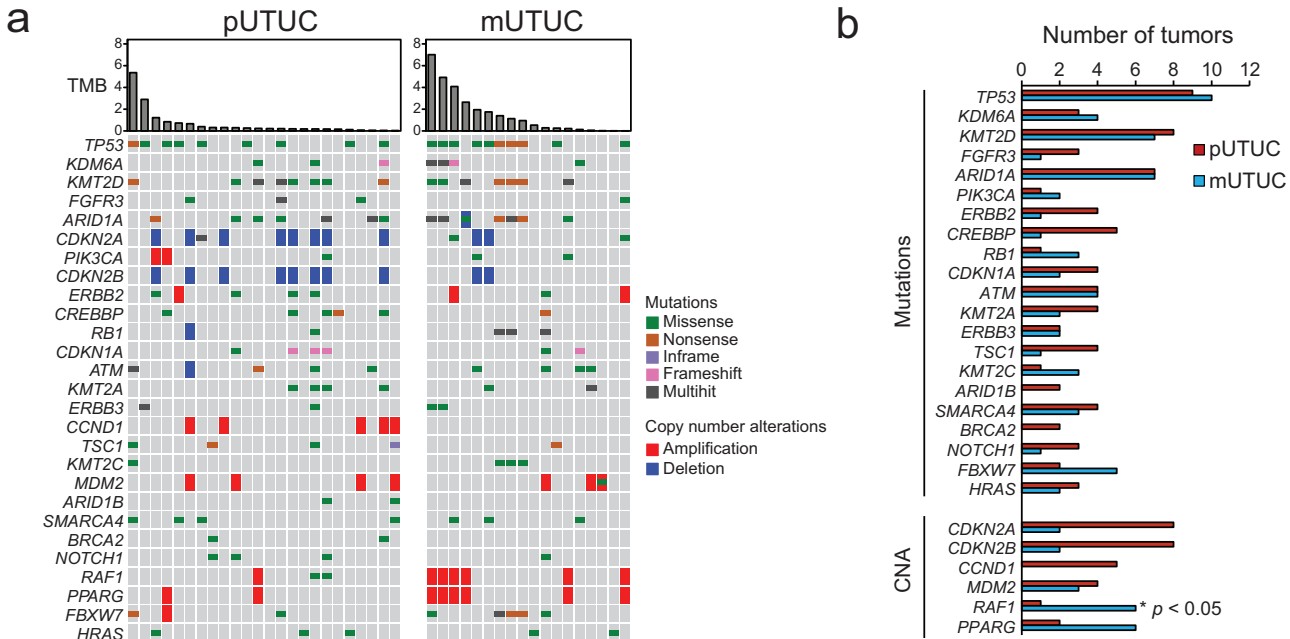

**Fig. 2 | Genomic landscape of primary and metastatic UTUC. a** Comparison of genomic alterations between primary and metastatic UTUC. **b** Comparison of genomic alteration frequencies between primary (red bars) and metastatic UTUC (blue bars) for genes frequently altered in our cohort. Source data are provided as a Source Data file.

Keratin 5, GATA3, E-cadherin), immune cells (CD3, CD16, CD68), among others such as functional or stromal markers present in various cell types (Supplementary Table 3). Upon inspection, IMC™ images generally revealed structured boundaries between tumor and stromal compartments (Fig. 5a, b). For a quantitative description of tumor and tumor-associated phenotypes, we segmented a total of 340,798 single cells across all images and, using the marker intensity as a quantitative measure of epitope abundance, built a joint space reflecting the phenotypic similarity between cells using the Uniform Manifold Approximation and Projection method (UMAP) (Fig. 5c). In this space, the tumor cells were readily distinguishable from the remaining cells by the basal marker Keratin 5 or the expression of either the luminal marker GATA3 or other keratin proteins (Fig. 5d). The remaining cells formed the immune and stromal compartments of the tumor, of which the most numerous were cancer-associated fibroblasts, macrophages, CD4+ and CD8+ T-cells, and endothelial cells (Fig. 5d, e). We found that the samples classified as immune-inflamed by the RNA-seq-based classifier had a higher number of CD8+ T-cells and Tregs (Supplementary Fig. 6). Additional immune cell populations were identified (Fig. 5e). We identified a population of cells positive for CD11b, CD68, granzyme B, and PD-L1 and weakly positive for CD11c and PD-1. The number of infiltrating cells from this population was higher in the samples classified as immune-depleted than immune-inflamed (Supplementary Fig. 6), suggesting a potential role in immune suppression. The phenotype of this population overlaps with myeloid-derived suppressor cells, a heterogeneous group of immune cells from the myeloid lineage. We labelled this population broadly as immunosuppressive myeloid cells because our panel of markers were not sufficient for further characterization of this population. In the tumor compartment, we identified a wide range of tumor phenotypes expressing various combinations of these markers, including the expression of tumor-specific proteins (Fig. 5e). Upon closer inspection, we observed that the degree of plasticity as measured by the ratio of the expression of the basal marker KRT5 to the luminal marker GATA3 (KRT5/GATA3) per cell was largely conserved across primary and metastatic UTUC within the same patient (Fig. 5f). However, for one patient (WCM031), the primary and metastatic tumor cells had distinct phenotypes, with

the primary tumor having a basal phenotype with low CD8+ T-cell infiltration. In contrast, the metastatic sample has a luminal phenotype and approximately double the number of CD8+ T-cells (78.6 cells in primary vs. 134.5 cells in metastasis, Supplementary Fig. 7). These findings suggest that cancer immune-contexture phenotypes are heterogeneous between patients but often conserved between primary and metastatic tumors within individual patients with UTUC.

## Discussion

Here, we describe the genomic, transcriptomic, and immunophenotypic landscape of metastatic UTUC. Furthermore, we used multiplexed tissue imaging using Imaging Mass Cytometry™ to describe the intra- and inter-tumoral phenotypic heterogeneity of UTUC tumors and their microenvironment at the single-cell level. Genomic data from paired primary and metastatic UTUC tumors in small cohorts have been reported[6,21]. Here we describe single-cell imaging analysis coupled with transcriptomic data.

WES analysis from paired primary and metastatic UTUC revealed that metastatic tumors harbored genomic alterations not identified in the matched primary tumors. Two possibilities can be considered to account for this heterogeneity: (a) Clones harboring these additional alterations are resistant to perioperative chemotherapy. (b) Metastatic lineages arise from early branched evolution in the primary tumors. These possibilities were discussed in previous reports, which suggested that systemic spread is a very early event in cancer history and that chemotherapy selects clones harboring drug-resistant mutations in breast, colorectal, lung, and bladder cancers[13,22]. In our cohort, intrapatient heterogeneity of alterations in cancer-related genes was also observed in the chemotherapy-naïve cases (WCM052 and WCM068). However, the percentages of shared mutations between primary and matched metastases were higher than those with a history of chemotherapy before sampling. Chemotherapy potentially plays a role in increasing genomic heterogeneity by inducing mutations[13–15]. Previous genomic studies in small cohorts showed discordance of mutations between paired primary and metastatic UTUC[6,21]. We show that intrapatient genomic diversity in clinically targetable alterations is also common in UTUC. This finding is of clinical importance because

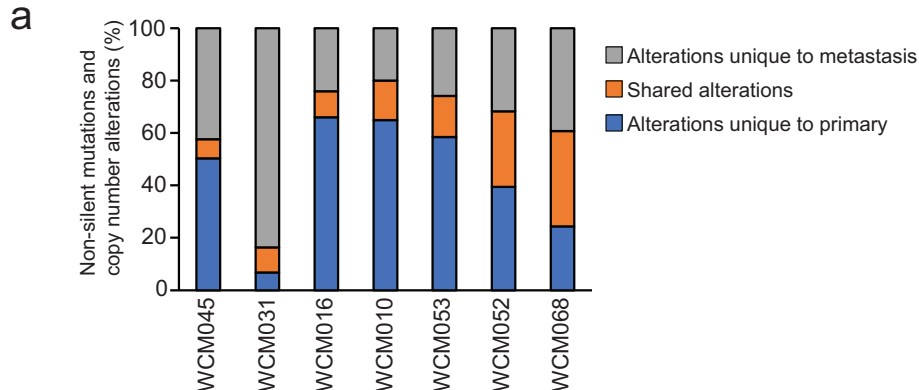

**Fig. 3 | Genomic heterogeneity between paired primary and metastatic UTUC.**
**a** Percentages of somatic mutations unique to primary (blue) and metastasis (gray) or shared between the matched tumors (orange). **b** Oncoprints representing somatic mutations and copy number alterations of cancer genes in primary (P) and matched metastatic (M) UTUC. Oncoprints are grouped per case. Chemotherapy treatment prior to tissue collection and a history of bladder cancer are shown on the top of each oncoprint. Alterations shown in purple indicate actionable altera-tions. Source data are provided as a Source Data file.

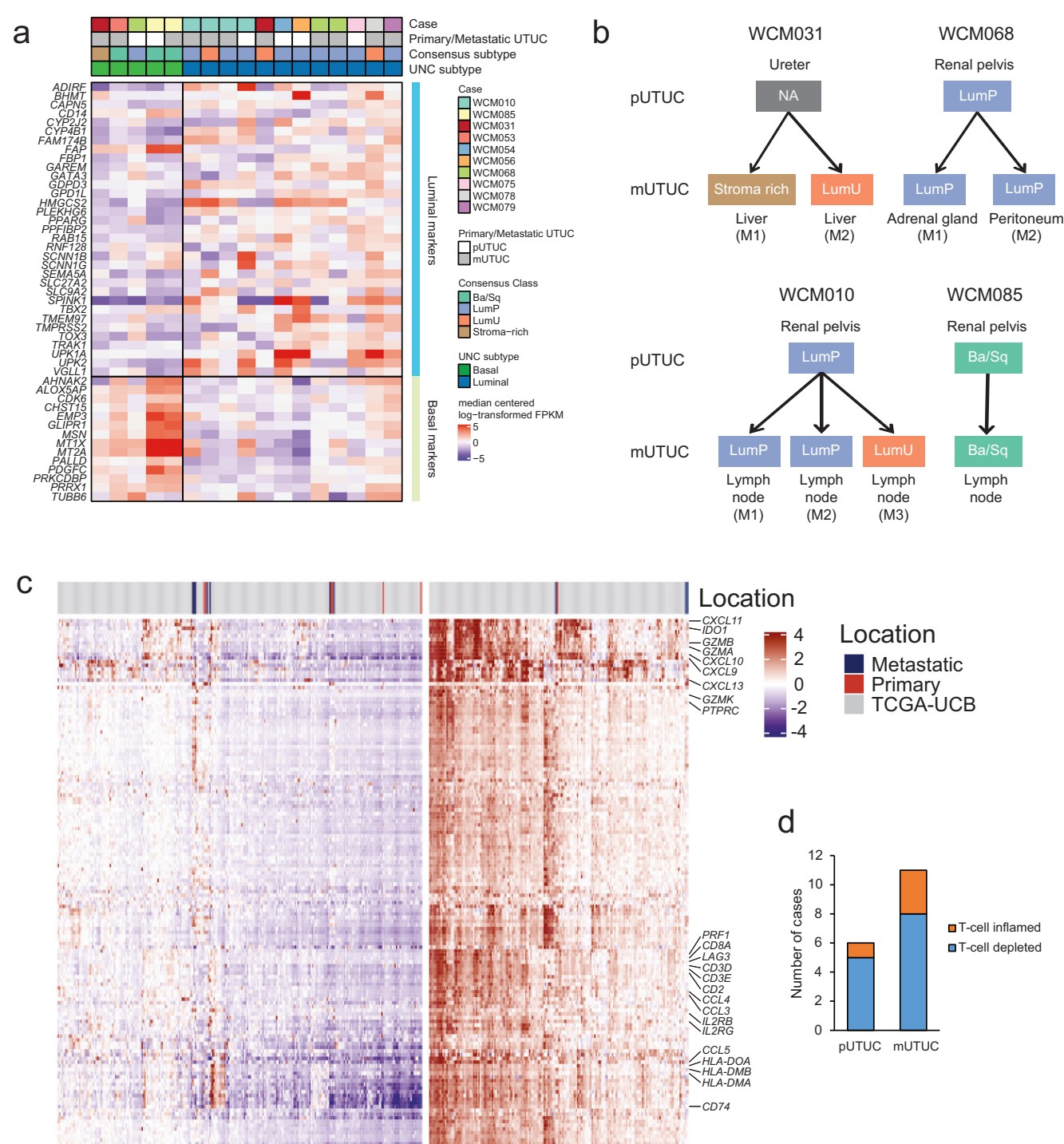

**Fig. 4 | Comparison of molecular subtype and T-cell inflammation classification in primary and metastatic UTUC. a** Gene expression heatmap of UTUC aligned by cases. The expression value is log-transformed and median centered for selected genes, for labeled gene sets. Assigned molecular classes are represented on top. The subtypes are labeled as luminal papillary (LumP), luminal non-specified (LumNS), luminal unstable (LumU) and basal/squamous (Ba/Sq). **b** Comparison of molecular subtypes between paired primary/metastatic and metastatic/metastatic UTUC. **c** Supervised consensus clustering of our UTUC and TCGA UCB tumors according to a 170-immune gene signature classifies tumors into T-cell depleted (with lower expression of classifier genes), and T-cell inflamed (with higher expression of classifier genes) clusters. **d** Breakdown of assigned classification for primary and metastatic UTUC. Source data are provided as a Source Data file.

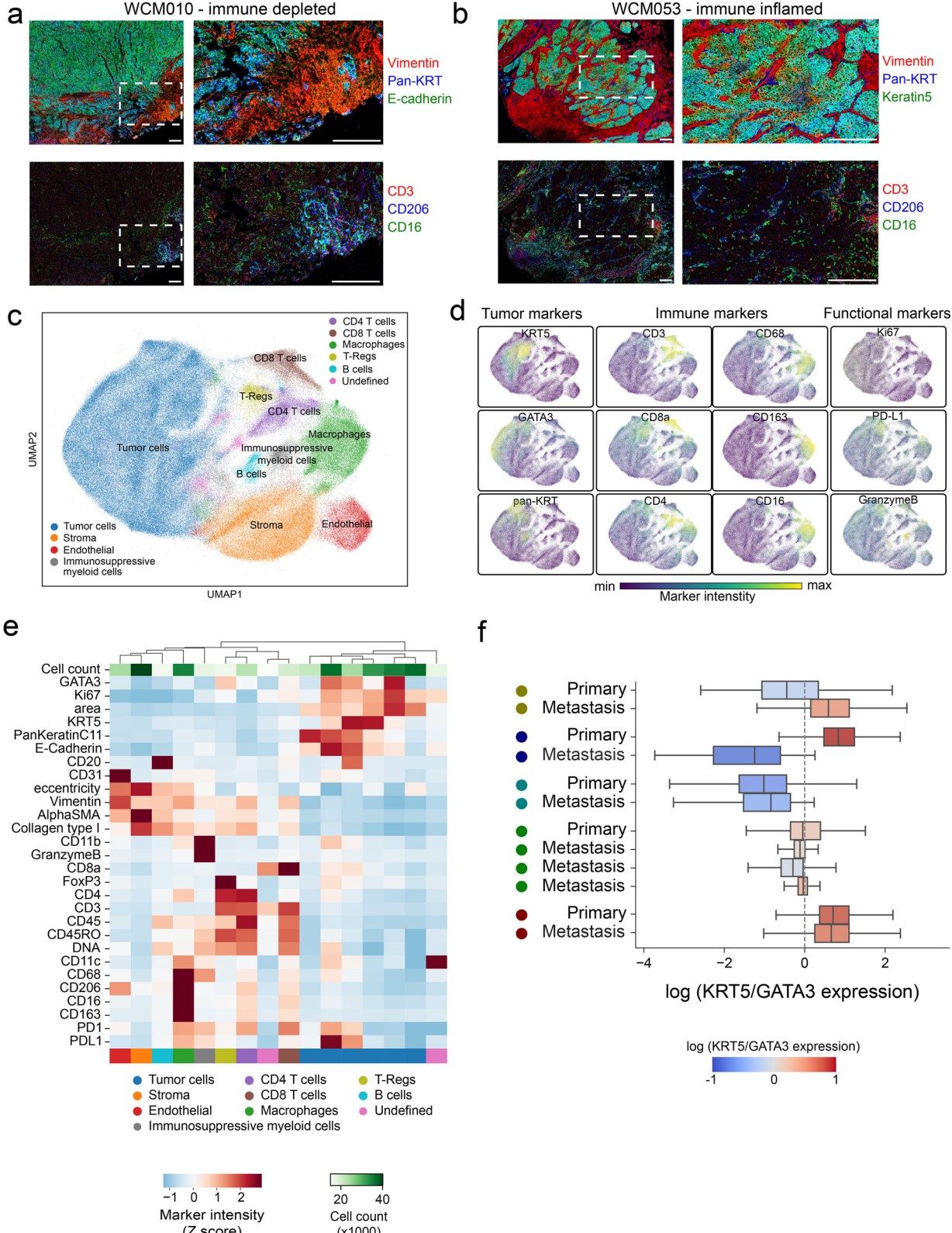

evaluating only the primary or specific metastatic sites could potentially miss therapeutic targets in patients with metastatic UTUC. However, we acknowledge that even the samples we collected in this study may not fully represent the entire tumor burden from each patient. We cannot rule out that some alterations detected only in metastases were present in unsampled regions of their paired primary tumors. Circulating cell-free tumor DNA liquid biopsies will likely complement tissue biopsy and thus capture the scope of genomic heterogeneity that single-lesion tumor biopsies may miss[23].

We observed intra-patient discrepancy of molecular subtypes within two patients (WCM01031 and WCM031) at the transcriptomic level using a recently published molecular classification system[17]. A recent study provided important findings which may explain the molecular basis of the molecular subtype shifts in urothelial

**Fig. 5 | Imaging Mass Cytometry™ reveals intra-patient concordance in molecular subtype plasticity between primary and metastatic UTUC tumors at the single cell level.** Representative images of IMC™ data from single experiment for a sample of the (**a**) immune-depleted ($n = 8$) or (**b**) immune-inflamed classes ($n = 4$). Red: tumor expressed, E-cadherin; Green: T-cell expressed CD3; Blue: DNA. Scale bars represent 200 microns horizontally. **c** UMAP representation of single-cells segmented in IMC™ data colored by the metacluster of origin. **d** UMAP representation as in c) but where each single cell is colored by the intensity of various markers in the panel. **e** Heatmap of clusters found in the IMC™ data (columns) and their average intensity in the markers. Clusters were aggregated into meta-clusters dependent on their ontogeny. **f** Log ratio of KRT5 to GATA3 expression for each single cell aggregated in boxplots for each sample in the IMC™ dataset (a total of 158,909 cells from $n = 12$ biologically independent samples). The vertical dotted line represents the level where the KRT5 expression is equal to the GATA3 expression. Centers, boxes, and whiskers indicate medians, quantiles, and minima/maxima, respectively. Source data are provided as a Source Data file.

carcinomas. Using a single-cell transcriptome and in vivo bladder cancer model, Sfakianos *et al.* revealed lineage plasticity in primary and metastatic urothelial cancers of the bladder[24]. The molecular subtype changes observed in our cohort may reflect the consequence of lineage plasticity during metastatic progression. In contrast, for cases WCM068 and WCM085, molecular subtypes were concordant between the matched primary and metastatic tumors. This observation is consistent with a recent study showing the molecular subtype remains stable during metastatic progression of urothelial carcinoma[25]. However, we are aware that the degree of intra-tumoral heterogeneity in expression-based molecular subtypes may be underestimated due to the sampling bias resulting from a single tumor biopsy. Intra-tumoral and intra-individual molecular heterogeneity has been shown in UC in previous studies[26–28]. The primary-metastatic concordance of molecular subtypes within each patient is of potential clinical relevance because molecular subtype membership is currently explored as a biomarker for predicting treatment response and overall survival[29,30]. A single snapshot biopsy does not reflect the full biological spectrum of evolving metastatic UTUC.

Interestingly, the expression of immune-related genes was relatively stable between primary and metastatic UTUC tumors. Accurate profiling of the immune microenvironment is clinically relevant because the extent of tumor-infiltrating T-lymphocytes is positively associated with the efficacy of immune checkpoint inhibitors[31]. Furthermore, the concordance of the immune phenotype within each patient in our study suggests that evaluating a single tissue site provides a useful assessment of the local immune microenvironment of metastatic UTUC tumors. These results have important implications for predicting the efficacy of immune checkpoint inhibitors.

Multiplexed imaging using IMC™ allowed us to comprehensively profile cancer cell phenotypes and the surrounding microenvironment in primary and metastatic UTUC, revealing the extensive heterogeneity between patients but conserved between primary and metastatic samples of the same individual in all but one patient. While this analysis includes a limited set of protein markers compared with RNA-seq, it allowed us to characterize the spatial heterogeneity of cancer, stromal and immune cells UTUC evolution at the single-cell level. Furthermore, this technique validated transcriptome-based predictions of the immune-inflamed or depleted microenvironment. This spatially resolved imaging technique generated additional insights not only at the level of quantifying the density of cellular components of the immune compartment, such as CD8+ T-cells, but also pinpointing their distribution, including the degree of infiltration into the tumor mass compared to their presence in the periphery. We identified a group of cells that we labeled immunosuppressive myeloid cells as they were positive for CD11b, CD68, granzyme B, and PD-L1 and weakly positive for CD11c and PD-1. PD-L1 expression suggests they have immunosuppressive properties overlapping with Myeloid-derived suppressor cells, a heterogeneous group of immune cells from the myeloid lineage[32–34]. The higher number of these cells in the immune-depleted tumors is consistent with their immunosuppressive activity. These cells also expressed CD11c, the most widely used defining marker for dendritic cells. Another possibility is that these cells are granulocytes since CD11b is also a marker of granulocytes and granzyme B is reportedly a potential marker of tumor-infiltrating neutrophils[35].

Examining HLA-DR expression would be needed to confirm whether the CD11c-positive cells in this subset are dendritic.

Our study integrates genomic, transcriptomic, and multiplexed spatial single-cell protein expression analyses from matched primary and metastatic UTUC tumors. The cohort size of 28 patients with UTUC is relatively small, in part due to the rarity of UTUC. This poses a challenge for generalizing the results to all metastatic UTUCs. Further validation studies using larger cohorts would be required. A second limitation is that only bulk tissue was sequenced from each tumor. Single bulk tumor sampling might underestimate intra-tumoral heterogeneity (ITH), which is now considered a key obstacle to success in cancer treatment because ITH is the molecular basis of metastatic dissemination and therapy resistance[36,37]. Multi-site sampling and single-cell RNA sequencing could generate additional insights into the degree of intratumoral ITH in metastatic UTUC. Finally, WES doesn't cover the entire genome, and silent mutations were not called by our clinically oriented analytical pipeline[38]. Whole-genome sequencing-based studies will further expand our knowledge of molecular underpinnings of metastatic UTUC.

Altogether, our genomic, transcriptomic, and phenotyping analysis of primary and metastatic UTUC brings to light the range of mutational and phenotypic diversity between and within individual patients, revealing a broadly conserved phenotypic tumor landscape against the backdrop of genetic evolution seen during metastasis. This integrated characterization of UTUC informs the targeted and immune therapeutic strategies that maximize efficacy in patients with metastatic UTUC.

## Methods

### Patient enrollment and tissue acquisition
Patients with primary and metastatic UTUC were prospectively enrolled at Weill Cornell Medicine in an institutional review board (IRB)–approved Research for Precision Medicine Study (WCM IRB No. 1305013903) with written informed consent. Retrospective tissue samples were retrieved and studied under the protocol for Comprehensive Cancer Characterization by Genomic and Transcriptomic Profiling (WCM IRB No. 1007011157). Tumor tissue from biopsies and nephroureterectomy specimens was collected from 44 patients diagnosed with high-grade urothelial carcinoma. WES data from 7 primary tumor samples presented in this manuscript have already been published[7]. Tumor DNA for WES was obtained from fresh frozen or formalin-fixed, paraffin-embedded tissue. Samples were selected based on pathologic diagnosis according to standard guidelines for UTUC[2,39]. Small cell carcinoma was excluded from our cohort. Pathological review by study pathologists (B.D.R., F.K., J.M.M.) confirmed the diagnosis and determined tumor content.

### DNA extraction and WES
We employed a clinical-grade WES assay (EXaCT-1), which is a test approved by the Department of Health at New York State (NYS-DOH ID#43032), to detect somatic mutations, indels and copy number alterations (CNA), as well as tumor mutational burden (TMB) and microsatellite instability (MSI)[38]. WES was performed on each patient's tumor/matched germline DNA pair. After macrodissection of target lesions, tumor DNA was extracted from either formalin-fixed, paraffin-embedded (FFPE) or cored OCT-cryopreserved tumors using the

Promega Maxwell 16 MDx (Promega, Madison, WI, USA). Germline DNA was extracted from blood, buccal mucosa, and normal lung and lymph node tissue using the same method. A minimum of 200 ng of DNA was used for WES. DNA quality was determined by TapeStation Instrument (Agilent Technologies, Santa Clara, CA) and was confirmed by real-time PCR before sequencing. Sequencing was performed using Illumina HiSeq 2500 (2 × 100 bp). A total of 21,522 genes were analyzed with an average coverage of 85× using Agilent HaloPlex Exome (Agilent Technologies, Santa Clara, CA).

## WES data processing pipeline

All the sample data were processed through the computational analysis pipeline of the Institute for Precision Medicine at Weill Cornell Medicine and NewYork-Presbyterian (IPM-Exome-pipeline)[38]. Raw read quality was assessed with FASTQC. Short reads were then aligned to GRC37/hg19 reference using BWA. The alignment quality of the aligned BAM files is obtained by calculating several metrics related to the average coverage and capture rate by calculating how many aligned reads fall within a capture region in the Agilent HaloPlex Whole Exome kit. Our capture rate is determined by the percent of mapped reads found overlapping any capture region in the Agilent HaloPlex Whole Exome kit and the total number of mapped reads of any given sample. High-quality capture rates range from ~80-95%. Average coverage is computed by calculating the average number of reads overlapping a capture region in the Agilent HaloPlex Whole Exome kit. Typically, the average coverage of a sample ranged from 80-100X (Supplementary Table 4). The tumor purity estimate is computed using CLONET[40]. Pipeline output includes segment DNA copy number data, somatic copy-number aberrations (CNAs), and putative somatic single-nucleotide variants (SNVs), as described in this section (Supplementary Data 1)[41].

## Detection of somatic SNVs

SNVs were identified in the paired tumor-normal samples using MuTect2, Strelka, VarScan, and SomaticSniper, and only the SNVs identified by at least two mutation callers were retained. Indels were identified using Strelka and VarScan, and only those identified by both tools were retained. Somatic variants were filtered using the following criteria: (a) read depth for both tumor and matched normal samples was ≥30 reads, (b) the variant allele frequency (VAF) in tumor samples was ≥10% and >5 reads harboring the mutated allele, (c) the VAF of matched normal was ≤1% or there was just one read with mutated allele. The variants were annotated using Oncotator (version 1.9); the dbSNPs amongst the mutation calls, unless also found in the COSMIC database, were filtered out. For the IPMs samples, the promiscuous mutation calls, previously identified internally as artifacts for Haploplex, were also excluded from the final list of mutations. Pathogenicity and actionability for each mutation and CNA were determined by the OncoKB database[16]. TMB was calculated as the number of mutations divided by the number of bases in the coverage space per million. TMB status (high vs. low) was determined using a urothelial cancer-specific threshold which our group recently reported[10].

## Detection of somatic copy number alterations

For somatic copy number alterations, the number of aligned reads for the capture regions in the Agilent HaloPlex Whole Exome Kit was calculated in both the tumor sample and matched control sample. Capture regions with total coverage of <100 reads in both the tumor and matched control samples are filtered out. For the remaining capture regions, read counts are normalized in both the tumor sample and the matched control sample by the total number of reads aligned in the tumor sample and the matched control sample, respectively. Then the ratio of the normalized read counts in the tumor sample and the normalized read count in the control sample is calculated. These capture regions are then ordered karyotypically and sorted by

genomic coordinates to help segment our capture regions according to the log2 value of the ratio of normalized read counts of the tumor sample and control sample in a biologically meaningful way. The normalized ratios of these bins were segmented using the Circular Binary Segmentation algorithm implemented in the R package DNAcopy. The algorithm outputs segments where every capture region found within these segments is represented by the same log2 value. This log2 value indicates whether the segment has DNA copy number amplification or DNA copy number deletion. Segments with a log2 value > 1 to are amplified, and segments with a log2 value < −0.5 are categorized as deleted. We then took the segments called by the algorithm and annotated these segments by RefSeq genes whose transcription start and end sites overlap with the genomic coordinates assigned to these segments.

## Computational MSI analysis

MSI was detected by the MSI sensor computational tool. This tool quantifies MSI in paired tumor–normal genome sequencing data and reports the somatic status of corresponding microsatellite sites in the human genome[11]. MSIsensor score was calculated by dividing the number of microsatellite-unstable by the total number of microsatellite-stable (MS) sites detected. The cut-off for defining MSI-high (MSI-H) versus MS stable (MSS) samples was 3.5 (MSI-H > 3.5, MSS < 3.5)[11].

## RNA extraction, RNA sequencing, and data analysis

RNA-seq and data processing were performed using the following procedures. Briefly, RNA was extracted from frozen material for RNA-seq using Promega Maxwell 16 MDx instrument (Maxwell 16 LEV simplyRNA Tissue Kit (cat. # AS1280)). Specimens were prepared for RNA sequencing using TruSeq RNA Library Preparation Kit v2 or riboZero. RNA integrity was verified using the Agilent Bioanalyzer 2100 (Agilent Technologies). cDNA was synthesized from total RNA using Superscript III (Invitrogen). Sequencing was then performed on GAII, HiSeq 2000, or HiSeq 2500 as paired-ends[42]. All reads were independently aligned with STAR_2.4.0f1[43] for sequence alignment against the human genome sequence build hg19, downloaded via the UCSC genome browser (http://hgdownload.soe.ucsc.edu/goldenPath/hg19/bigZips/), and SAMTOOLS v0.1.19[44] for sorting and indexing reads. Cufflinks (2.0.2)[45] was used to estimate the expression values (FPKMS), and GENCODE v23[46] GTF file for annotation. Rstudio with R (v3.6.1) was used for the statistical analysis and the generation of figures.

## Molecular subtyping

The log-transformed expression data was used to infer molecular subtypes using a recently published classification system as implemented in the consensusMIBC R package[17]. Default parameters were set except for minCor representing a minimal threshold for best Pearson's correlation (minCor = 0.15). The consensus classification implements a nearest centroid method and Pearson's correlation and classifies samples into 6 molecular classes: Luminal Papillary (LumP), Luminal Non-Specified (LumNS), Luminal Unstable (LumU), Stroma-rich, basal/Squamous (Ba/Sq), Neuroendocrine-like (NE-like).

## T-cell inflammation classification analysis

The previously published T-cell inflammation gene signature was used to classify tumors into T-cell inflamed or depleted[7]. The expression data, quantified as FPKMs, was obtained for the EIPM UTUC patients, of which RNA-seq was available (n = 17). The FPKMs for the primary tumors were obtained from the GDC/TCGA bladder cohort (TCGA BLCA, n = 414). The genes from the signature were selected for expression-based supervised clustering. The FPKMs were log-transformed and median centered and partitioning around medoids (PAM) algorithm was applied to cluster the transformed expression data. This led to the identification of two broad clusters: one with

higher expression of the signature genes was labeled as T-cell inflamed, while that with low expression of signatures genes was labeled as T-cell depleted.

## Deconvolution using Cibersort

We used Cibersort to estimate the relative abundance of 22 leukocytes in the tumor microenvironment for each tumor in our cohort ($n = 17$)[19]. Cibersort was applied to the RNAseq expression profiles (FPKMs) without quantile normalization (as recommended by the developers of this method). We used the default LM22 signature as the reference matrix and the default 'relative' mode for result normalization. From the results, we filtered out the cell types for which the Cibersort estimates were not significant ($p$ values < 0.05). Cibersort provides an empirical p-value testing the null hypothesis that a particular sample does not contain any of the 22 cell types. Two samples were excluded because the empirical $p$ value for the deconvolution was over 0.05.

## GSEA Pathway analysis

We applied the single sample gene set enrichment analysis (GSEA) to the 17 UTUC RNAseq expression profiles using the Hallmark pathways from the Molecular Signatures Database (MSigDB)[47–49]. The enrichment scores from the ssGSEA analysis were standardized using the z-score normalized across all samples for each pathway. The p-values for statistically significant pathways between the primary and metastatic tumors were calculated using the Wilcoxon signed-rank test.

## Imaging Mass Cytometry™

Antibodies were conjugated in BSA and Azide-free format using the MaxPar X8 multimetal labeling kit (Standard BioTools™) per the manufacturer's protocol. These antibodies were tested on control tissues such as lymph node and tonsil to validate the staining pattern as verified by our pathologist. Freshly cut 4-micron thick FFPE tissue sections were stored at 4 ℃ for a day before staining. Slides were first incubated for 1 hour at 60 °C on a slide warmer, followed by dewaxing in fresh CitriSolv (Decon Labs) twice for 10 minutes, rehydrated in descending series of 100%, 95%, 80%, and 75% ethanol for 5 minutes each. After 5 minutes of MilliQ water wash, slides were treated with antigen retrieval solution (Tris-EDTA pH 9.2) for 30 minutes at 96oC, cooled to room temperature (RT), washed twice in TBS, and blocked for 1.5 hours in SuperBlock Solution (Thermo Fischer). Overnight incubation occurred at 4oC with the prepared antibody cocktail containing the metal-labeled antibodies (Supplementary Table 3). The next day, slides were washed twice in 0.2% Triton X-100 in PBS and twice in TBS. DNA staining was performed using Intercalator-Iridium in PBS solution for 30 minutes in a humid chamber at room temperature, followed by a washing step in MilliQ water and air drying. The Hyperion instrument was calibrated using a tuning slide to optimize the sensitivity of the detection range. Hematoxylin and Eosin (H&E) stained slides were used to guide the selection of regions of interest in order to obtain representative regions. All ablations were performed with a laser frequency of 200 Hz. Tuning was performed intermittently to ensure the signal detection integrity was within the detectable range.

## Analysis of Imaging Mass Cytometry™ data

Imaging Mass Cytometry™ data were preprocessed by the following steps. First, image data was extracted from MCD files acquired with the Hyperion instrument. Hot pixels were removed using a fixed threshold. The image was amplified two times. Gaussian smoothing was applied, and from each image, a square 500-pixel crop was saved as an HDF5 file for image segmentation. Segmentation of cells in the image was performed with *ilastik* (version 1.3.3)[50] by manually labeling pixels as belonging to one of three classes: nuclei – the area marked by a signal in the DNA and Histone H3 channels (Supplementary Table 3); cytoplasm – the area immediately surrounding the nuclei and overlapping with signal in cytoplasmic channels; and

background – pixels with low signal across all channels. Ilastik learns from these sparse labels by training a Random Forest classifier using features present in the images. Features used were the Gaussian Smoothing with kernel widths of 1 and 10 pixels, Hessian of Gaussian Eigenvalues with kernel 3.5 and 10 pixels, and Structure of Tensor Eigenvalues with kernel of 10 pixels. The outputs of prediction are class probabilities for each pixel which were used to segment the using DeepCell version 0.8.2)[51] with the *MultiplexSegmentation* pre-trained model.

To identify cell types in an unsupervised fashion, we first quantified the intensity of all samples in each segmented cell, not overlapping image borders. Channels with "functional" markers Ki67, PD-1, PD-L1, and Granzyme B were not used for downstream cell type identification but only for visualization. In addition, for each cell, we computed morphological features such as the cell area, perimeter, the length of its major axis, eccentricity, and solidity using the skimage.measure.regionprops_table function (version 0.18.1)[51]. Cells with a solidity value of 1 (perfectly round cells) were excluded from the analysis. Using Scanpy (version 1.7.1)[52], we log-transformed the quantification matrix, and Z-scored values per image, capping the signal at −3 and 3, followed by global feature centering and scaling. The batch was removed with Combat (Scanpy implementation), and features were scaled again. Principal Component Analysis was performed, and we computed a neighbor graph on the PCA latent space using batch-balanced k-nearest neighbors (bbknn) (version 1.4.0)[53]. We computed a Uniform Manifold Approximation and Projection (UMAP)[54] embedding (umap package, version 0.4.6) with a gamma parameter of 25, and clustered the cells with the Leiden algorithm[55] with resolution 0.5 (leidenalg package, version 0.8.3).

## Statistical analyses

A two-sided Mann–Whitney test was used for statistical tests to check for significant differences between the two distributions. The two-sided Fisher's exact test was applied to determine whether the deviations between the observed and the expected counts were significant. We used a *p-value* threshold of 0.05.

## Reporting summary

Further information on research design is available in the Nature Portfolio Reporting Summary linked to this article.

# Data availability

The raw sequence data that support the findings of this study are available in the database of Genotypes and Phenotypes (dbGaP) under the accession number: phs001087.v3.p1. For IMC™, the pre-processed.tiff files are available at https://doi.org/10.5281/zenodo.5719187. Source data are provided with this paper.

# Code availability

Source code used to analyze IMC™ data are available at https://github.com/ElementoLab/utuc-imc and deposited on Zenodo: https://doi.org/10.5281/zenodo.10230334[56].

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

## Acknowledgements

This work was supported by the Cornell Center for Immunology Core Facilities Seed Grant (J.M.M., K.O., O.E., B.M.F.) and by the Caryl and Israel Englander Institute for Precision Medicine. A.F.R. is supported by an NCI T32CA203702 grant. B.M.F. was supported by the Gellert Family-John P. Leonard, MD Research Scholarship in Hematology and Medical Oncology. For technical support, the authors thank Bing He, Ruben Diaz, and Leticia Dizon from the Center for Translational Pathology of the Department of Pathology and Laboratory Medicine at Weill Cornell Medicine.

## Author contributions

Initiation and design of the study: K.O., B.M.F. and J.M.M. Subject enrollment, sample, pathology review, and clinical data collection: K.O., D.N., D.P., A.V., K.S.S., J.M., S.B., B.D.R., F.K., S.T.T., M.M.O., B.M.F. and J.M.M. Lab data collection and analysis: K.O., D.W. and B.M.F. Imaging mass cytometry: H.R. and statistical and bioinformatic analyses: K.O., A.F.R., B.B., K.W.E., R.B., E.F., A.S. and O.E. Supervision of research: B.M.F. and J.M.M. Writing of the first draft of the manuscript: K.O., B.M.F. and J.M.M. All authors contributed to the writing and editing of the revised manuscript and approved the manuscript.

## Competing interests

B.M.F. Consulting or Advisory Role Astrin Biosceince, Inc, Natera, Guardant, Janssen, Boston Gene, Astra Biosciences, Gilead, Merck, Immunomedics, QED therapeutics and has received research support for Weill Cornell from Eli Lilly, received patent royalties from Immunomedics and Gilead Sciences and received honoraria from Urotoday. O.E. holds equity in OneThree Biotech, Volastra Therapeutics, Owkin, Champions Oncology, Pionyr Immunotherapeutics, Harmonic Discovery and Freenome. S.T.T. has received research funding (to Weill Cornell Medicine) from Sanofi, Astellas, Janssen, Amgen, Lilly, Genentech, BMS, AstraZeneca, Bayer, Merck, Abbvie, Clovis, Seagen, Novartis, Gilead, POINT Biopharma, Ambrx, and has received honoraria from Sanofi, Astellas, Janssen, Bayer, Eisai, Abbvie, Tolmar, Seagen, Amgen, Clovis, Pfizer, Novartis, Genomic Health, POINT Biopharma, Blue Earth, Aikido, Telix, Convergent, EMD Serono, Myovant, Merck, Daiichi Sankyo. The remaining authors declare no competing interests. No patents have been filed or are related to this manuscript.

## Additional information

[1]Department of Pathology and Laboratory Medicine, Weill Cornell Medicine, New York, NY 10065, USA. [2]Englander Institute for Precision Medicine, Weill Cornell Medicine, New York, NY 10021, USA. [3]Department of Physiology and Biophysics, Weill Cornell Medicine, 1300 York Avenue, New York, NY 10065, USA. [4]Institute for Computational Biomedicine, Weill Cornell Medicine, 1305 York Avenue, New York, NY 10021, USA. [5]CeMM Research Center for Molecular Medicine of the Austrian Academy of Sciences, Lazarettgasse 14 AKH BT 25.3, 1090 Vienna, Austria. [6]Department of Medicine, Division of Hematology and Medical Oncology, Weill Cornell Medicine, New York, NY 10065, USA. [7]Sandra and Edward Meyer Cancer Center at Weill Cornell Medicine, New York, NY 10065, USA. [8]Departments of Cell and Developmental Biology, Weill Cornell Medicine, New York, NY 10065, USA. [9]These authors contributed equally: Bishoy M. Faltas, Juan Miguel Mosquera. ✉e-mail: bmf9003@med.cornell.edu; jmm9018@med.cornell.edu

