## [Peer Review File · Nature Communications]

REVIEWER COMMENTS

Reviewer #1 (Remarks to the Author): Expert in bladder cancer genomics and pathology

The paper by K. Ohara et al. describes the results of genomic, transcriptomic, and single-cell imaging mass cytometry analyses of upper track primary and metastatic urothelial carcinoma. The communication is complementary to the previously published papers on the same subject but provides an interesting advancement of our knowledge on evolution of molecular profile of upper track bladder carcinoma to metastatic disease.

My specific comments are as follows:

1. More detailed description of whole exome sequencing processing pipeline as well as the results is recommended i.e. depth of sequencing of individual samples or in average for primary and metastatic tumor, total number of silent, non-silent mutations and the specific nucleotide substitutions in individual samples should be provided.

2. Validation of sequencing results by PCR-based Sanger sequencing of selected genes and/or calculating the sensitivity of WES by comparing DNA-based sequencing data to RNASeq data similar to those used in bladder cancer TCGA reports should be provided.

3. The analyses of mutagenesis signature in primary versus metastatic tumors will add value to the analyses.

4. In view of the evidence that there is an evolutionary pattern of mutational change in primary and metastatic tumor the analyses of positive/negative as well as neutral selection patterns of individual mutations could be of interest.

5. More in depth quantitative analysis of immune signature e.g. CIBERSORT will provide additional insights to the tumor microenvironment concept.

6. RNASeq analyses is superficial and expanding it to the comparative pathway analyses between primary and metastatic tumors complemented by quantitative assessment such as GSEA will add value to this communication.

MINOR COMMENTS

- P-value in Figure 5G is needed to validate the significance of change versus no change among primary and metastatic tumors.
- Supplementary table with a list of all mutations, their nucleotide positions, nucleotide substitutions and amino acid change is needed.

Reviewer #2 (Remarks to the Author): expert in bladder cancer, immune tumour microenvironment, and mass cytometry

In their study, Ohara and colleagues nicely describe the tumor heterogeneity at the single-cell level across patients with primary and metastatic UTUC. The authors integrated a combination of experimental platforms ranging from WES, bulk RNAseq, and IMC.

Main findings include "actionable" genomic alterations are in fact quite different between the primary and metastatic tumors within UTUC patients. This is novel and may present problems for predicting the best treatment strategies. The authors perform very elegant analyses. However, the study fails to drive anything "actionable". In the end, tumors are highly heterogeneous but there are some shared phenotypes between primary and metastatic tumors intrapatient.

Unfortunately, it's the opinion of this reviewer that this study lacks in breadth beyond establishing heterogeneity in the TME and does not provide enough advance to the field of cancer immunology.

Reviewer #3 (Remarks to the Author): Expert in urothelial carcinoma and immune tumour microenvironment

Authors investigated the molecular characteristics of mUTUC using multi-omics study. The study was well designed and highly informative to understand the basic characters of UTUC. UTUC is embryologically different from bladder cancer, but the studies about molecular characteristics have been relatively fewer reported than bladder cancer. The study could provide the comprehensive understanding of UTUC. However, the current study had inevitable two problems.

First is that "Could the samples harvested synchronously from primary cancer or metastatic tissues represent the whole tumor characteristics due to intratumoral, inter-metastatic organ or metachronous heterogeneity?" Please describe the authors' opinion about that.

Second is that “Could small patients samples represent the evolution of multi-omics characteristics of UTUC?” I understand the samples in the current study were very important. But the analysis of the results may give the false information due to small numbers. Please describe the discussion session about that.

Reviewer #4 (Remarks to the Author): Expert in cancer immunogenomics and Imaging Mass Cytometry

Major comments:

The authors have employed IMC to provide an (immune) cell spatial profiling of UTUC's. In general the IMC data seems underexplored as no models of interaction between different subsets have been applied and the panel applied by the authors does not really allow them to dig deeper into specific subsets within major lineages.

Also, the IMC approach was only applied in 14 samples derived from 6 patients. Are the data points presented in figure 5F corresponding to individual images? This is not correct from a statistical standpoint to compare amount of cells per images, between groups that have been defined per patient.

Finally, it would be useful for the authors to include the performance of the individual antibodies as supplementary material as this information is of great value for other groups performing IMC. For instance, groups are struggling to have CD80/CD86 immunodetection on IMC but the authors did not reveal whether this has been successful (it was not included in the heatmap).

Other comments:

In the abstract, what is meant with “the evolution of cancer cell, immune cell, and stromal cell markers using mass cytometry”, in particular the word evolution?

I am confused by the use of reference 19 to support the imaging mass cytometry approach.

Why were CD45/CD86 included in the same channel?

Reviewer #1 (Remarks to the Author):

The paper by K. Ohara et al. describes the results of genomic, transcriptomic, and single-cell imaging mass cytometry analyses of upper tract primary and metastatic urothelial carcinoma. The communication is complementary to the previously published papers on the same subject but provides an interesting advancement of our knowledge on evolution of molecular profile of upper tract bladder carcinoma to metastatic disease.

My specific comments are as follows:

1. More detailed description of whole exome sequencing processing pipeline as well as the results is recommended i.e. depth of sequencing of individual samples or in average for primary and metastatic tumor, total number of silent, non-silent mutations and the specific nucleotide substitutions in individual samples should be provided.

Response: In accordance with the Reviewer's comments, we have added a detailed description of our whole exome sequencing processing pipeline, especially on the detection of somatic single-nucleotide variants and somatic copy number alterations. Quality metrics for WES of each sequenced sample have been provided in Supplementary table 4.

2. Validation of sequencing results by PCR-based Sanger sequencing of selected genes and/or calculating the sensitivity of WES by comparing DNA-based sequencing data to RNASeq data similar to those used in bladder cancer TCGA reports should be provided.

Response: We appreciate the Reviewer's valuable comments. To validate the WES results, we compared WES results with the corresponding targeted gene panel data. We found two samples which were interrogated by both WES and a targeted panel and confirmed a druggable *FGFR3* mutation detected by OncoPrint in one sample (WCM081_P) and a druggable *PIK3CA* mutation by a custom made 50 gene panel in another sample (WCM057_M1), respectively. This fact has been clearly described in the Results section of the revised manuscript on page 5.

3. The analyses of mutagenesis signature in primary versus metastatic tumors will add value to the analyses.

4. In view of the evidence that there is an evolutionary pattern of mutational change in primary and metastatic tumor the analyses of positive/negative as well as neutral selection patterns of individual mutations could be of interest.

Response: We completely agree with the Reviewer's comments that analyses of mutagenesis signature and evolutionary pattern of somatic mutations are important. Unfortunately, we are unable to perform these analyses because our dataset is not best suited for them. There are mainly two reasons. The first is that our mutation calling pipeline (EXaCT-1) is a clinical grade pipeline focusing on drivers and other potentially clinically relevant mutations. Therefore, silent mutations are not called in our pipeline. The second is that we used the Agilent HaloPlex Whole Exome kit, which doesn't cover the entire exonic regions of the genome. Because of these limitations, we are unable to

perform a robust mutational signature analysis or evaluate positive, negative and neutral selection of individual mutations. Future studies using whole genome sequencing would be required and are currently underway but are beyond the scope of the current manuscript. We acknowledge this as one of the limitations of our study and based on your suggestion, we have added a line at the end of the Discussion (page 13).

5. More in depth quantitative analysis of immune signature e.g. CIBERSORT will provide additional insights to the tumor microenvironment concept.

Response: As suggested by the Reviewer, we have now profiled tumor infiltrating immune cells by CIBERSORT. CIBERSORT showed similar immune profiles between primary and metastatic UTUC. Next, we focused on infiltration of T cells since both primary and metastatic UTUC were T-cell depleted. CIBERSORT demonstrated that activated memory CD4 T cells and Gamma Delta T cells were low compared to other T cell subtypes in the tumor microenvironment of UTUC. These facts have been clearly described in the Results section (Supplementary Figure 5).

6. RNASeq analyses is superficial and expanding it to the comparative pathway analyses between primary and metastatic tumors complemented by quantitative assessment such as GSEA will add value to this communication.

Response: We appreciate the Reviewer's suggestion. To address this point, we have now performed GSEA analysis. GSEA didn't reveal any significant differences of gene sets between primary and metastatic UTUC. We have added this result in the Result section of in the revised manuscript (Supplementary Figure 4).

MINOR COMMENTS

- *P-value in Figure 5G is needed to validate the significance of change versus no change among primary and metastatic tumors.*

Response: We agree with the Reviewer's comments that the p-value is important to show the significance of the difference. However, we are unable to add p-value in Figure 5g from a statistical standpoint related to Reviewer #4's comment below. Because the data points in the boxplots in Figure 5g represent the expression ratios (KRT5/GATA3) of cancer cells per individual tumor, the data points in the same boxplot are considered not independent. This makes p-values inappropriate.

- *Supplementary table with a list of all mutations, their nucleotide positions, nucleotide substitutions and amino acid change is needed.*

Response: We have provided a list of all non-synonymous mutations and copy number alterations detected by WES in Supplementary Table 5. The Raw genomic data is also available in the database of Genotypes and Phenotypes (dbGaP).

Reviewer #2 (Remarks to the Author):

In their study, Ohara and colleagues nicely describe the tumor heterogeneity at the single-cell level across patients with primary and metastatic UTUC. The authors integrated a combination of experimental platforms ranging from WES, bulk RNAseq, and IMC.

Main findings include "actionable" genomic alterations are in fact quite different between the primary and metastatic tumors within UTUC patients. This is novel and may present problems for predicting the best treatment strategies. The authors perform very elegant analyses. However, the study fails to drive anything "actionable". In the end, tumors are highly heterogeneous but there are some shared phenotypes between primary and metastatic tumors intrapatient.

Unfortunately, it's the opinion of this reviewer that this study lacks in breadth beyond establishing heterogeneity in the TME and does not provide enough advance to the field of cancer immunology.

Response: We thank the Reviewer for these comments. To our knowledge, there are no previous reports performing whole exome and transcriptome data of paired primary and metastatic UTUC. Moreover, we show, for the first time, we use highly multiplexed imaging mass cytometry data that allowed us to analyze the TME spatial profiles of UTUC at the single-cell level. We believe that this integrated multidimensional omics dataset will provide insight into UTUC biology and provide an important resource for other researchers working on this rare tumor type.

Reviewer #3 (Remarks to the Author):

Authors investigated the molecular characteristics of mUTUC using multi-omics study. The study was well designed and highly informative to understand the basic characters of UTUC. UTUC is embryologically different from bladder cancer, but the studies about molecular characteristics have been relatively fewer reported than bladder cancer. The study could provide the comprehensive understanding of UTUC. However, the current study had inevitable two problems.

First is that "Could the samples harvested synchronously from primary cancer or metastatic tissues represent the whole tumor characteristics due to intratumoral, inter-metastatic organ or metachronous heterogeneity?" Please describe the authors' opinion about that.

Response: We agree with the Reviewer's comments that existence of intratumoral, inter-metastatic organ and synchronous/metachronous heterogeneity should be considered. We acknowledge that a bulk tumor might include a mixed population of tumor cells harboring distinct molecular features and that the bulk tumor sampling that we employed in this study might underestimate the heterogeneity. This is a limitation of our study which we have now acknowledged in the discussion section. Our analysis using imaging mass cytometry has provided integrative spatial protein expression at a single-cell level. However, further studies using multi-regional sampling or single-cell DNA sequencing would help

shed light on a more precise profiling of tumor heterogeneity. We have added this point as a limitation in the Discussion section of the revised manuscript (Page 12).

Second is that “Could small patients samples represent the evolution of multi-omics characteristics of UTUC?” I understand the samples in the current study were very important. But the analysis of the results may give the false information due to small numbers. Please describe the discussion session about that.

Response: We acknowledge that a small sample size is a limitation of this study. Validation studies using larger cohorts and functional experiments would be required in order to reveal UTUC biology. We have now addressed this limitation in the Discussion section of the revised manuscript (Page 12).

Reviewer #4 (Remarks to the Author):

Major comments:

The authors have employed IMC to provide an (immune) cell spatial profiling of UTUC's. In general the IMC data seems underexplored as no models of interaction between different subsets have been applied and the panel applied by the authors does not really allow them to dig deeper into specific subsets within major lineages. Also, the IMC approach was only applied in 14 samples derived from 6 patients. Are the data points presented in figure 5F corresponding to individual images? This is not correct from a statistical standpoint to compare amount of cells per images, between groups that have been defined per patient.

Response: We thank the Reviewer for this valuable feedback. As the Reviewer pointed out, the data points shown in Figure 5f correspond to individual images. We agree that comparison of data points from individual images/tumor is inaccurate. Therefore, we deleted p-values from Figure 5f and changed the description of the result.

Finally, it would be useful for the authors to include the performance of the individual antibodies as supplementary material as this information is of great value for other groups performing IMC. For instance, groups are struggling to have CD80/CD86 immunodetection on IMC but the authors did not reveal whether this has been successful (it was not included in the heatmap).

Response: Antibodies were validated with appropriate controls using immunohistochemistry (IHC), as presented on the manufacturer's datasheet. Custom conjugated clones were internally validated using IHC and verified by a board-certified pathologist. Moreover, the results of IMC performed at our institution have been published at multiple peer-reviewed journals (Cold Spring Harb Mol Case Stud. 2022 Apr 28;8(3):a006151., Nature. 2021 May;593(7860):564-569). We have now mentioned these details in the Reporting summary (page 3). Regarding CD80/CD86, the antibodies were not employed in this study. We apologize for any confusion from this error. We have now deleted CD80 and CD86 from the list of IMC antibodies (Supplementary Table 3).

Other comments:

In the abstract, what is meant with “the evolution of cancer cell, immune cell, and stromal cell markers using mass cytometry”, in particular the word evolution?

Response: We have now clarified this sentence in the abstract which now reads “We performed spatially-resolved single-cell analysis of cancer, immune, and stromal cells to understand the evolution of primary to metastatic UTUC.”

I am confused by the use of reference 19 to support the imaging mass cytometry approach.

Response: We apologize for this mistake. We have now corrected the wrong citation to the correct one: Chang Q, Ornatsky OI, Siddiqui I, Loboda A, Baranov VI, Hedley DW. Imaging Mass Cytometry. Cytometry A. 91:160-169. doi:10.1002/cyto.a.23053 (2017).

Why were CD45/CD86 included in the same channel?

Response: Apologies for the confusion. As mentioned above, CD80/CD86 were not used in this study, so we have now revised Supplementary Table 3, that is a list of IMC antibodies employed for this study to reflect this update.

REVIEWER COMMENTS

Reviewer #1 (Remarks to the Author):

Authors adequately responded to all of my comments.

Reviewer #2 (Remarks to the Author):

In their revisions, K. Ohara and colleagues have, in fact, nicely addressed many of the original criticisms by the other reviewers.

Additionally, the authors should be congratulated on a very nicely descriptive analysis comparing upper track primary and metastatic urothelial carcinomas. This is a largely understudied area of research and very important to better understand.

I still believe that from a fundamental perspective, this study lacks the ability to translate any meaningful differences between primary and metastatic tumors. Partly, the authors did not identify any big differences and partly because the small sample size, albeit appreciating how challenging it is to acquire clinical specimens for these analyses.

I do believe that the authors have addressed most of the other reviewers' criticisms as well as nicely provided limitations of the study. The topic is of direct interest to the readership of the journal. I would be happy to see this manuscript published on Nature Communications.

Reviewer #4 (Remarks to the Author):

I thank the authors for addressing my points. I still have the opinion that there are very limited insights revealed by the IMC approach. It adds little more than revealing a potential association with the RNAseq data. But even then, some claims are unsupported. For instance, the median number of CD8+ T cells is the same between inflamed and non-inflamed groups. It's striking the authors did not highlight instead the dendritic/MDSC group as this would be the most striking difference.

In any case, I do not agree that based on the markers presented, the authors can classify those cells as dendritic/MDSC. Those cells are positive for CD11b, granzyme B(!), PD-1 (!), PD-L1. Actually, their "undefined" type is most likely to be dendritic cells due to the strong expression of CD11c but HLA-DR would be required to confirm this.

In sum, I do not think the IMC data significantly adds to the paper and that it deserves a full image. Details like the ones I mentioned should also be addressed

Reviewer #1 (Remarks to the Author):

Authors adequately responded to all of my comments.

Response: We wish to thank the reviewer for this comment.

Reviewer #2 (Remarks to the Author):

In their revisions, K. Ohara and colleagues have, in fact, nicely addressed many of the original criticisms by the other reviewers.

Additionally, the authors should be congratulated on a very nicely descriptive analysis comparing upper track primary and metastatic urothelial carcinomas. This is a largely understudied area of research and very important to better understand.

I still believe that from a fundamental perspective, this study lacks the ability to translate any meaningful differences between primary and metastatic tumors. Partly, the authors did not identify any big differences and partly because the small sample size, albeit appreciating how challenging it is to acquire clinical specimens for these analyses.

I do believe that the authors have addressed most of the other reviewers' criticisms as well as nicely provided limitations of the study. The topic is of direct interest to the readership of the journal. I would be happy to see this manuscript published in Nature Communications.

Response: We appreciate the reviewer's comment.

Reviewer #4 (Remarks to the Author):

I thank the authors for addressing my points. I still have the opinion that there are very limited insights revealed by the IMC approach. It adds little more than revealing a potential association with the RNAseq data. But even then, some claims are unsupported. For instance, the median number of CD8+ T cells is the same between inflamed and non-inflamed groups. It's striking the authors did not highlight instead the dendritic/MDSC group as this would be the most striking difference.

In any case, I do not agree that based on the markers presented, the authors can classify those cells as dendritic/MDSC. Those cells are positive for CD11b, granzyme B(!), PD-1 (!), PD-L1. Actually, their "undefined" type is most likely to be dendritic cells due to the strong expression of CD11c but HLA-DR would be required to confirm this.

In sum, I do not think the IMC data significantly adds to the paper and that it deserves a full image. Details like the ones I mentioned should also be addressed

Response: We agree with the Reviewer's comments that a set of markers we used are not enough to define some cell types. We acknowledge that in the interpretation of the concordance between single cell RNA defined cell populations and those defined based

on flow cytometry marker genes can be challenging. Our initial rationale for designating as dendritic/MDSC was that positive for CD11b, CD68, granzyme B and PD-L1, and weakly positive for CD11c and PD-1. We acknowledge that this designation was too specific without adequate information. Therefore, in response to the Reviewer's comments, we replaced the "dendritic/MDSC" designation with the term "immunosuppressive myeloid cells" throughout the manuscript. We revised the figures 5c and 5e to reflect this change. We have hypothesized that their potential immunosuppressive activity was consistent with our observation that the number of the immunosuppressive myeloid cells in immune-depleted was higher than immune-inflamed. Furthermore, we have moved panel 5f which includes the different cell types based on the markers we used to the Supplementary materials. Also, we have highlighted the "immunosuppressive myeloid cells" instead of CD8+ T-cells in Result section (page 9).

Regarding undefined cells strongly positive for CD11c, we acknowledge that HLA-DR is one of crucial markers for dendritic cells and should be analyzed to confirm whether CD11c-positive cells are dendritic cells. Unfortunately, HLA-DR was not included in a panel of antibodies we used. Repeating the entire panel to include HLA-DR would be a significant endeavor and since defining this population is not the focal point of our manuscript, we hope that our re-designation of the population with a more appropriate albeit more general label would address this issue. We now acknowledge this as one of the limitations of our study and has been added in the Discussion (page 13-14).

We have discussed these points with Dr. Madhu M. Ouseph, who is a board-certified hematopathologist at Weill Cornell Medicine, and prepared the revised manuscript with him and added him as a coauthor.

REVIEWERS' COMMENTS

Reviewer #4 (Remarks to the Author):

I thank the authors for addressing my comments. I still stand by the opinion that the IMC data does not contribute substantially to the paper, except perhaps to support the author's statement that there is no obvious heterogeneity between primary and metastatic samples albeit more samples would be required to confirm this.

I suggest that the authors remove the following statement: "We found that the samples classified as immune-inflamed by the RNA-seq-based classifier had a higher number of Tregs, effectively validating a 170-gene RNA-seq-based classifier for immune characterization of UTUC (Supplementary Figure 6f)." I don't see how elevated Tregs validate their 170 gene RNA signature.

What the authors described as MDSCs can also be granulocytes. I agree that this population should be highlighted as potentially playing a role in immune suppression and this should be further investigated in subsequent studies.

Reviewer #4 (Remarks to the Author):

I thank the authors for addressing my comments. I still stand by the opinion that the IMC data does not contribute substantially to the paper, except perhaps to support the author's statement that there is no obvious heterogeneity between primary and metastatic samples albeit more samples would be required to confirm this.

I suggest that the authors remove the following statement: "We found that the samples classified as immune-inflamed by the RNA-seq-based classifier had a higher number of Tregs, effectively validating a 170-gene RNA-seq-based classifier for immune characterization of UTUC (Supplementary Figure 6f)." I don't see how elevated Tregs validate their 170 gene RNA signature.

What the authors described as MDSCs can also be granulocytes. I agree that this population should be highlighted as potentially playing a role in immune suppression and this should be further investigated in subsequent studies.

Response:

We appreciate your valuable comments and agree with them.

First, we would like to apologize for replacing Supplementary Figure 6 showing comparison of structural cells between immune-depleted and immune-inflamed samples. When we were reviewing data to prepare for this revision and an author checklist, we found a data input mistake in T-cell inflammation status for one tumor. Consequently, we have corrected all box plots in Supplementary Figure 6. Though the correction doesn't impact the conclusion, the higher infiltration of CD8⁺ T-cells and T-regs in immune-inflamed samples have become more pronounced. We have added the correct result on CD8⁺ T-cells and Tregs in Result section (page 9).

We agree with your opinion that the 170 gene RNA signature doesn't validate elevated Tregs. Therefore, the statement pointed out has been modified as follows: "We found that the samples classified as immune-inflamed by the RNA-seq-based classifier had a higher number of CD8⁺ T-cells and Tregs, ~~effectively validating a 170-gene RNA-seq-based classifier for immune characterization of UTUC (Supplementary Figure 6f)~~".

Regarding the possibility that the cell population labeled as MDSC is granulocytes, we acknowledge that CD11b is also a marker of granulocytes and granzyme B is reportedly a potential marker of tumor-infiltrating neutrophils. We discussed the point in the Discussion section (page 13).